# Convolutional Neural Network for Individual Identification Using Phase Space Reconstruction of Electrocardiogram

**DOI:** 10.3390/s23063164

**Published:** 2023-03-16

**Authors:** Hsiao-Lung Chan, Hung-Wei Chang, Wen-Yen Hsu, Po-Jung Huang, Shih-Chin Fang

**Affiliations:** 1Department of Electrical Engineering, Chang Gung University, Taoyuan 333, Taiwan; 2Biomedical Engineering Research Center, Chang Gung University, Taoyuan 333, Taiwan; 3Neuroscience Research Center, Chang Gung Memorial Hospital, Linkou, Taoyuan 333, Taiwan; 4Department of Neurology, Cardinal Tien Hospital Yung Ho Branch, New Taipei City 234, Taiwan

**Keywords:** ECG biometric, individual identification, phase space reconstruction, convolutional neural network

## Abstract

Electrocardiogram (ECG) biometric provides an authentication to identify an individual on the basis of specific cardiac potential measured from a living body. Convolutional neural networks (CNN) outperform traditional ECG biometrics because convolutions can produce discernible features from ECG through machine learning. Phase space reconstruction (PSR), using a time delay technique, is one of the transformations from ECG to a feature map, without the need of exact R-peak alignment. However, the effects of time delay and grid partition on identification performance have not been investigated. In this study, we developed a PSR-based CNN for ECG biometric authentication and examined the aforementioned effects. Based on a population of 115 subjects selected from the PTB Diagnostic ECG Database, a higher identification accuracy was achieved when the time delay was set from 20 to 28 ms, since it produced a well phase-space expansion of P, QRS, and T waves. A higher accuracy was also achieved when a high-density grid partition was used, since it produced a fine-detail phase-space trajectory. The use of a scaled-down network for PSR over a low-density grid with 32 × 32 partitions achieved a comparable accuracy with using a large-scale network for PSR over 256 × 256 partitions, but it had the benefit of reductions in network size and training time by 10 and 5 folds, respectively.

## 1. Introduction

Electrocardiogram (ECG) has been widely used for physiological monitoring and clinical diagnosis. ECG also discloses specific individual characteristics, which can be used for biometric authentication with an advantage in security because it is acquired from a living body. Other people cannot easily obtain it from the daily environment, unlike a fingerprint, which can be taken from the devices that have been touched by fingers. In addition, the acquired ECG during telecare monitoring provides a biometric authentication to know the identity of the patient [1]. On the other hand, ECG data taken for biometric authentication are beneficial for the individual’s daily healthcare recording.

A typical ECG is composed of P, QRS complex, T, and U waves which are generated by a sequence of atrial depolarization, ventricular depolarization, and repolarization. Because each person has a respective cardiac conduction system and projection of the generated electrical activity on body surface, the waveform of the recorded ECG varies from person to person. Early ECG biometrics use several morphological parameters, such as widths, amplitudes, slopes, and intervals on the basis of the characteristic waves [2,3,4,5,6,7]. However, some morphological information remains undisclosed by these reduced features, but it is useful for individual identification. A good quality of the acquired ECG signals is essential, especially for the early ECG biometric methods. The identifications of small-amplitude characteristics such as P, Q, and S waves are commonly affected by the presence of noise contaminations or when the characteristic waves are difficult to discern.

Direct use of waveform [5,8,9] or transformed coefficients [7,10,11] as input features solves the aforementioned problem. An alternative approach is to directly transform ECG to a three-dimensional phase-space trajectory using a time-delay technique [12,13]. Pattern match [5,9,13] and morphological difference [7,8,9,10,11,12,13] are mostly used to quantify similarity and dissimilarity between two ECGs. Even though the waveform-based methods do not need the detections of the small-amplitude waves, QRS detection is still needed for the segmentation of a cardiac cycle for individual identification. Digital filtering is commonly used to attenuate low-frequency baseline wandering and high-frequency noises [14]. Advanced methods using multi-resolution wavelet analysis [15,16] and kalman filtering [17,18] are also proposed to enhance ECG waveforms for QRS detection and extract ECG features.

Convolutional neural networks (CNNs) outperform the aforementioned methods in terms of feature extraction because convolutions can produce discernible features from ECG. In the convolutional layers, the network weights are derived using machine learning on the basis of multiple individuals’ data. Several transformations of raw ECG are used to generate input features for the CNNs. A simple method is to put single-beat ECG samples in a vector, then one-dimensional (1D) convolutional networks are used to produce distinguishing features for a succeeding connected network to identify individuals [19]. The other methods transform the 1D ECG signals to two-dimensional (2D) images by plotting each ECG beat as an individual grayscale image (time–amplitude representation) [20], decomposing each ECG beat on various scales using continuous wavelet transform (time–scale representation) [21], and embedding each ECG beat onto a 2D phase-space image using a time-delay technique [22]. In addition, multi-beat ECG is also transformed to an image by stacking beat-aligned amplitudes [23,24] or merging multiple phase-space trajectories [25]. The 2D CNNs are therefore used to extract discernible features for succeeding connected networks to identify individuals.

Phase space reconstruction (PSR) uses a time-delay technique to reconstruct the phase-space trajectory of a signal. The signal is embedded into a multi-dimensional phase space by a plot of each sample *x*(*t*) versus its respective samples after fixed time delays *x*(*t* + *τ*), *x*(*t* + 2*τ*), …, *x*(*t* + (*m* − 1)*τ*), which was proposed initially for the purpose of disclosing nonlinear dynamics of the signal on the basis of chaos physics [26]. PSR was firstly used to analyze ECG for examining whether ventricular fibrillation is an instance of deterministic chaos [27]. Moreover, the reconstructed phase portraits reveal various morphologies associated with normal and pathological ECGs. Thus, several studies employed the PSR with *m* = 2 to detect QRS complex [28], recognize ventricular extrasystoles [29], and classify the type of ventricular arrhythmia [30] because the 2D reconstruction displays a concise phase space trajectory and lends itself more readily to feature extraction. In addition, PSR transforms the temporal patterns of the ECG signal (P, QRS, and T waves) to specific spatial loops in phase space without the need of characteristic wave detections, extending its application to individual identification [12,13,22,24].

The value of time delay determines the morphology of the reconstructed phase-space trajectory and whether the ECG characteristics are sufficiently expanded in the phase space. However, the time delay is empirically determined in the majority of studies without an investigation to explore the effect of various time delays on the accuracy of the ECG biometric authentication. On the other hand, the phase space is partitioned by a series of intersecting vertical and horizontal lines. The grid partition generates a 2D image array to record the absence or presence of the reconstructed trajectory in each partitioned square. The density of the grid partition determines whether the generated portrait image is fine-grained or coarse-grained, which may affect the accuracy of individual identification, but it has not been reported in previous works. In this study, we propose a novel ECG biometric model which utilizes a deep CNN to learn and identify individuals’ patterns over the reconstructed phase portrait. We also investigate the aforementioned effects on the accuracy of the proposed ECG biometric model using the underlying combinations of various time delays (*τ* = 2, 8, 16, 24, and 36 ms) and various-density grid partitions (16 × 16, 32 × 32, 64 × 64, 128 × 128, and 256 × 256) on the basis of the data from the PTB Diagnostic ECG Database, with a purpose of determining an optimal time delay and a lower-complexity model with the maintenance of identification performance.

The rest of this paper is organized as follows. The method section describes ECG biometric data, phase space reconstruction of ECG, and ECG biometric network. In the result section we present the accuracies of individual identification on the basis of phase-space features using various time delays and various-density grid partitions. In the discussion section we talk over beat-to-beat variation, the effects of time delay and grid partition on identification accuracy, and the scale-down of the proposed network.

## 2. Materials and Methods

### 2.1. ECG Biometric Data

This study used the data from the PTB Diagnostic ECG Database established by the Physikalisch-Technische Bundesanstalt (PTB), National Metrology Institute of Germany [31]. The PTB database contains 549 records from 290 subjects, with different time durations of around 2 min. Only 52 subjects are healthy, 148 subjects suffer from myocardial infarction, and 18 have cardiomyopathy or heart failure. Each signal is digitized at 1 kSa/s (kilo samples per second) with a resolution of 16 bits. ECG recordings which had a time duration of less than 100 s or were seriously corrupted by noises or artifacts were excluded, so that the included recordings contained more than 125 heart beats with less noise contaminations for network training and testing. Accordingly, a total of 115 ECG recordings covering 17 healthy subjects, 81 subjects with myocardial infarction, and 15 subjects with cardiomyopathy or heart failure were selected to develop our ECG biometric model.

Compared to lead augmented vector left/right (aVL/aVR) that is electrical voltage difference between left/right arm electrode and a combination of right/left arm electrode and left leg electrode, lead I ECG is defined as the voltage difference between the electrodes on left and right arms, which can also be obtained through the fingers of both hands and is the most practical measurement for ECG biometric authentication. Therefore, we chose lead I ECG as an input for the ECG biometric model in this study.

### 2.2. Data Processing and Phase Space Reconstruction

Since a good quality of the ECG signals is important for recognizing individuals’ ECG patterns, we employed digital filtering to keep a high signal-to-noise ratio in the processed ECGs prior to the application to the ECG biometric model. The following two anti-causal Butterworth filters were used: (1) a high-pass filter with a cutoff frequency of 1 Hz to remove low-frequency baseline wander, and (2) a lowpass filter with a cutoff frequency of 40 Hz to suppress high-frequency noises. The segmentation of individual ECG signal into heartbeats was performed using the following steps:(1)QRS complex was detected using a Python ECG QRS detector [32] based on the Pan-Tompkins algorithm [33].(2)Each detected QRS was further verified to exclude wrong detections and ventricular premature contraction beats.(3)For each inclusive heartbeat, the preceding 35% and the succeeding 65% of the samples around the QRS fiducial point in a cardiac cycle were extracted.

After the aforementioned processing, 125 available heartbeats could be extracted for network training and testing.

We used PSR to delineate ECG’s morphological features by the following steps: the ECG was upsampled by a factor of 10 using Gaussian interpolation, then normalized to a series of values from 0 to 1 by
$$x_{n}\left(t\right)=\frac{x\left(t\right)-x_{\min}}{x_{\max}-x_{\min}}$$
where *x*_max_ and *x*_min_ were the maximum and minimum of *x*(*t*). The normalized, upsampled ECG was embedded into a series of 2D vectors with time delay *τ*.
$$X\left(t\right)=\left[x_{n}\left(t\right)\;x_{n}\left(t+\tau \right)\right]$$

Plotting these vectors on a 2D coordinate plane produced a phase-space trajectory. As shown in Figure 1, the phase-space trajectory characterizes the morphological relationship of samples related to their delayed samples. QRS complex is transformed to an outer loop, which is the major loop in the phase space. T wave forms an inner loop and P wave, a minor loop. A phase portrait was finally generated as input features for ECG biometric network.

### 2.3. ECG-Biometric Network

A PSR-based CNN was developed for ECG biometric authentication. Single-cycle ECG was transformed to a phase portrait used as input features to the CNN. As listed in Table 1, five convolutional layers, modified from the AlexNet [34], were used to extract low- and high-level features from the phase portrait. The output of the last convolution layer was flattened and fed into a fully connected network, which contained two hidden layers and one output layer of 115 nodes corresponding to 115 individuals. The activation function was chosen as ReLu function in the hidden layers, where the output layer used softmax function to represent a categorical probability distribution.

### 2.4. Network Training and Testing

The proposed ECG biometric network was implemented in the Visual Studio Code Ver. 1.57.0 (Microsoft Corp., Redmond, WA, USA) using Python Ver. 3.8.10 (Python Software Foundation, Wilmington, DE, USA) and Pytorch Ver. 1.9.0 (Pytorch Org., USA). Networks were then trained and validated in a server computer with an eight-core Intel CoreTM i9-9900K CPU (Intel, Santa Clara, CA, USA) and a 24-GB Titan RTX GPU (Nvidia, Santa Clara, CA, USA) using CUDA technology.

Each ECG recording was partitioned into the following two parts: the former 100 heartbeats were included as training data, and the latter 25 heartbeats were used to validate the trained network. A total of 75 epochs were applied to train the network using an Adam optimization algorithm based on cross entropy loss function. To prevent overfitting and force the network to learn general and robust patterns from the data, a fraction of the hidden units in the fully connected network were randomly dropped at every iteration at a probability of 0.5. At every training epoch, the trained network was tested by the validation data. The accuracy was defined as the percentage of correct identifications among all validation data.

To examine the effect of time delay and grid partition on the accuracy of individual identification, a total of 50 combinations of ten time delays (*τ* = 2, 4, 8, 12, 16, 20, 24, 28, 32, and 36 ms) and five types of grids (16 × 16, 32 × 32, 64 × 64, 128 × 128, and 256 × 256 partitions) were used to generate phase portraits for individual identifications. To have all examinations be performed on the basis of the same network architecture, all phase portraits were interpolated to 2D images of the same size (256 × 256 pixels).

## 3. Results

Figure 2 shows the phase portraits transformed from the single-cycle ECG using various time delays over various-density grid partitions. Its QRS, T, and P waves are respectively transformed to a major loop, a secondary loop, and a minor loop on each portrait image, but no visible U-related loops are observed because of unapparent U wave in Figure 1. Using a small time delay (*τ* = 2 or 4 ms) narrows QRS loops and condenses P and T loops. When the time delay is set to 8, 12, and 16 ms, smooth, well-expanded QRS loops are produced. As the time delay is further increased, T and P loops are better expanded, but QRS loops are flattened (*τ* = 20 and 24 ms) and twisted (*τ* ≥ 28 ms). On the other hand, ECG phase portraits on the basis of a 256 × 256 grid partition disclose the most distinct phase-space trajectory. As the grid partition is reduced, phase-space trajectories becomes ambiguous, in particular, the parts attributed to P and T waves.

Figure 3a shows the accuracy of individual identification over the first 30 epochs on the underlying phase portraits embedded by various time delays. No matter which time delay was adopted, more training epochs were required to reach a saturated accuracy when a low-density grid partition was used compared to the use of high-density grid partition. That is, more epochs were required for learning discernible features from coarse-phase portraits. As shown in Figure 3b, more training epochs were also needed to reach a saturated accuracy when individual identification was performed on the basis of phase portraits embedded using a small time delay than those using a large time delay, no matter which density was adopted in grid partition. That is, more epochs were required for learning discernible features from poorly expanded phase portraits.

Figure 4 shows the average accuracy of individual identification on the underlying phase portraits using the combinations of five grid partitions (r = 16 × 16, 32 × 32, 64 × 64, 128 × 128, and 256 × 256 partitions) and ten time delays (*τ* = 2, 4, 8, 12, 16, 20, 24, 28, 32, and 36 ms). The representative accuracies are listed in Table 2. In the early training stage (epochs 16~30), identification performances were poor when a small time delay and low-density grid partition were used (97.05% at *τ* = 2 ms and r = 16 × 16 partitions) because of insufficient phase-space expansion and poor portrait details. Even though more training epochs (epochs 61~75) had been executed, its accuracy (98.14%) was still lower than those using a high-density grid partition (99.05% at *τ* = 2 ms and r = 256 × 256 partitions), large time delay (99.26% at *τ* = 36 ms and r = 16 × 16 partitions), or both (99.51% at *τ* = 36 ms and r = 256 × 256 partitions). The average accuracies are also plotted in Figure 5. Higher identification accuracies were achieved when the time delays were set to 20, 24 and 28 ms.

## 4. Discussion

The intervals between ECG characteristics, particularly between R and T, vary as heart period, creating shifts of the associated samples in the featured vector. Fortunately, the phase-space reconstruction maps ECG characteristics to specific loops in the phase space, which are less affected by interval variations. In addition, respiration-induced movements of the thoracic wall related to the heart typically cause beat-to-beat variations of QRS morphology, thereby producing scaling and shifting of phase-space loops [35]. Normalization of ECG amplitudes between 0 and 1 can fix the problem of QRS morphological variation, as shown in Figure 6a. Even so, some loop variations are still observed among consecutive normalized ECG beats, as shown in Figure 6b. In this study, we used 100 consecutive single-beat phase portraits from each individual to train the network. The existence of loop variations in the training data also provided a kind of data augmentation to increase the robustness of CNN to the loop variation.

PSR coverts morphological characteristics (temporal patterns) in an ECG signal into phase-space loops (spatial patterns). The converted spatial patterns are different among individuals, leading to good performances in ECG biometrics [12,13,22,25]. In particular, each spatial pattern has specific locations in the phase space, allowing 2D CNN to efficiently capture the phase-space fingerprints. The adopted time delay in these studies was chosen to produce a transformed phase-space trajectory that is as smooth as possible. In this study, we examined the effect of time delay on ECG biometrics. The use of a small time delay caused insufficient expansions of ECG characteristic waves, whereby more training epochs were required to achieve a saturated accuracy, but the accuracy was still lower than that achieved from using a large time delay. In addition, QRS loop was flattened or twisted, in addition to a better expansion of P and T waves, when the time delay was increased to 20 ms or larger. This study demonstrated a better identification performance when the time delays were set as 20, 24 and 28 ms. We regarded that this was because P, QRS, and T waves were well expanded in these configurations.

The PSR of ECG samples over the grid with a high number of partitions depicts a phase-space trajectory in more details, but it also increases the size of the generated portrait images; accordingly, the scale of the convolution layers should be enlarged to learn the portrait features. In this study, all phase portraits were interpolated to 256 × 256 pixels no matter how many partitions were used to generate ECG phase portrait; therefore, the achieved accuracy could be compared on the basis of the same-scale neural network. As a result, phase space expansion based on 16 × 16 partitions produced the lowest accuracy and needed more training epochs to reach a saturated accuracy. The use of 32 × 32 partitions improved this situation, and the obtained accuracy was close to those using 64 × 64, 128 × 128, or 256 × 256 partitions. A similar, scaled-down network, as illustrated in Table 3, was designed especially for portrait images with 32 × 32 pixels. Figure 7 shows the comparison of identification accuracy using the scaled-down network and original network which were designed for portrait images with 32 × 32 and 256 × 256 pixels, respectively, but the portrait images were derived on the basis of PSR over 32 × 32 partitions. The scaled-down network produced comparable accuracies to the original network. The scaled-down network could largely reduce the number of weights from 72,362,963 to 6,711,251, and the training time from 26 min 32 s to 5 min 37 s.

Pathological heartbeats caused by extra-systolic ventricular contractions are commonly different from the healthy heartbeats in the same individual, so they cannot be recognized as the individual’s pattern by the trained networks on the basis of healthy heartbeats. Fortunately, extra-systolic heartbeats commonly occur intermittently or interlaced (bigeminy or trigeminy). There are still healthy heartbeats that can be recognized by the trained network. That is why most ECG biometric authentication methods select healthy heartbeats to develop their biometric networks.

Table 4 shows the state of the art in ECG biometric authentication methods using two-dimensional convolutional neural networks. Two studies used three-layer CNN to extract discernible features from ECG images [22,25]. Two studies [20,21] and the present study used deeper neural networks, such as VGG [36], AlexNet [34], GoogleNet [37], and ResNet [38], to capture deep-learning ECG features. A study used the ensemble of two three-layer CNNs and a recurrent neural network to identify individuals’ ECGs [24]. As shown in a study by Byeon et al. [39], the accuracy of individual identification was slightly increased as the network went deeper in sequence of AlexNet, GoogleNet, and ResNet. Although, it is not easy to compare the accuracy of individual identification among various studies because of various data selections in different datasets. The proposed architecture on the basis of the AlexNet and PSR could achieve high accuracy in individual identification. The effect of network depth and architecture on individual identification over the ECG phase-space features can be further studied in future work.

## 5. Conclusions

PSR of ECG discloses individual trajectory patterns no matter which time delay was adopted. An appropriate time delay from 20 to 28 ms was suggested since it produced a good phase-space expansion of P, QRS, and T waves and a better performance in individual identification using the proposed CNN. In addition, using a high-density grid with 256 × 256 partitions depicted a detailed phase-space trajectory and yielded a high identification accuracy, but needed a large-scale neural network to deal with the augmented information. The use of a scaled-down network on the basis of PSR over a low-density grid with 32 × 32 partitions achieved a comparable accuracy with the benefit of reductions in network size and training time by 10 and 5 folds, respectively. In summary, temporal-to-spatial mapping through PSR directly produces 2D images with disclosed ECG characteristics, enabling an efficient learning of individual fingerprints by the 2D CNN. The use of the scaled-down CNN with appropriate time delay largely reduces the complexity of the model with a maintenance of identification performance, which is beneficial for practical implementation of the ECG biometric on authentication systems.

## Figures and Tables

**Figure 1 sensors-23-03164-f001:**
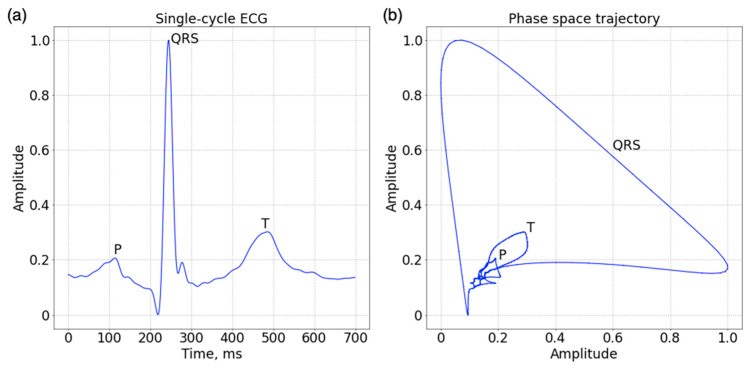
(**a**) A single-cycle ECG with P, QRS complex, and T waves. (**b**) The transformed phase portrait using a time delay of 20 ms.

**Figure 2 sensors-23-03164-f002:**
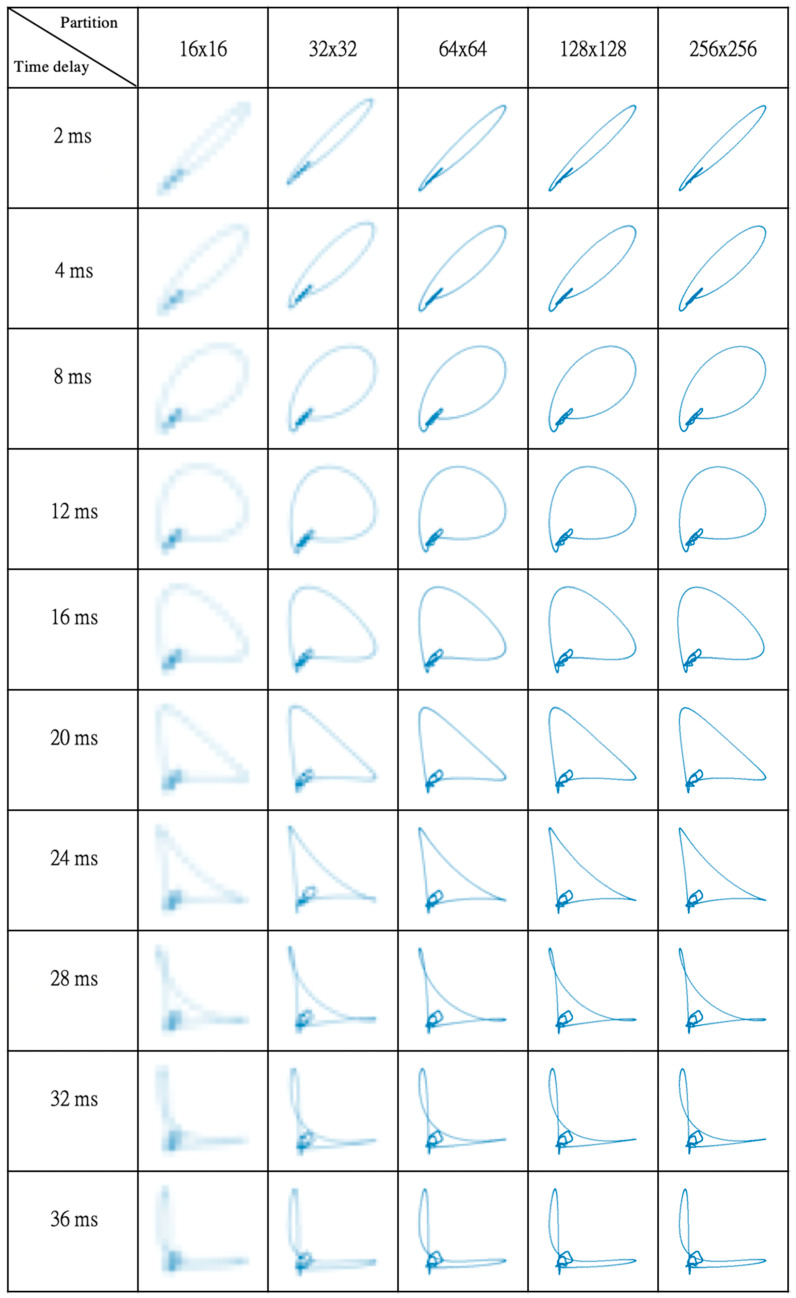
Phase portraits transformed from single-cycle ECG, using various time delays over various-density grid partitions. QRS complex, T, and P waves are transformed to a major outer loop, a secondary inner loop, and a minor loop, respectively, in each portrait. The distinctness of the phase portrait decreases as its grid partition is reduced from 64 × 64 to 16 × 16 partitions. For a visual comparison, all phase portraits are displayed with the same picture size.

**Figure 3 sensors-23-03164-f003:**
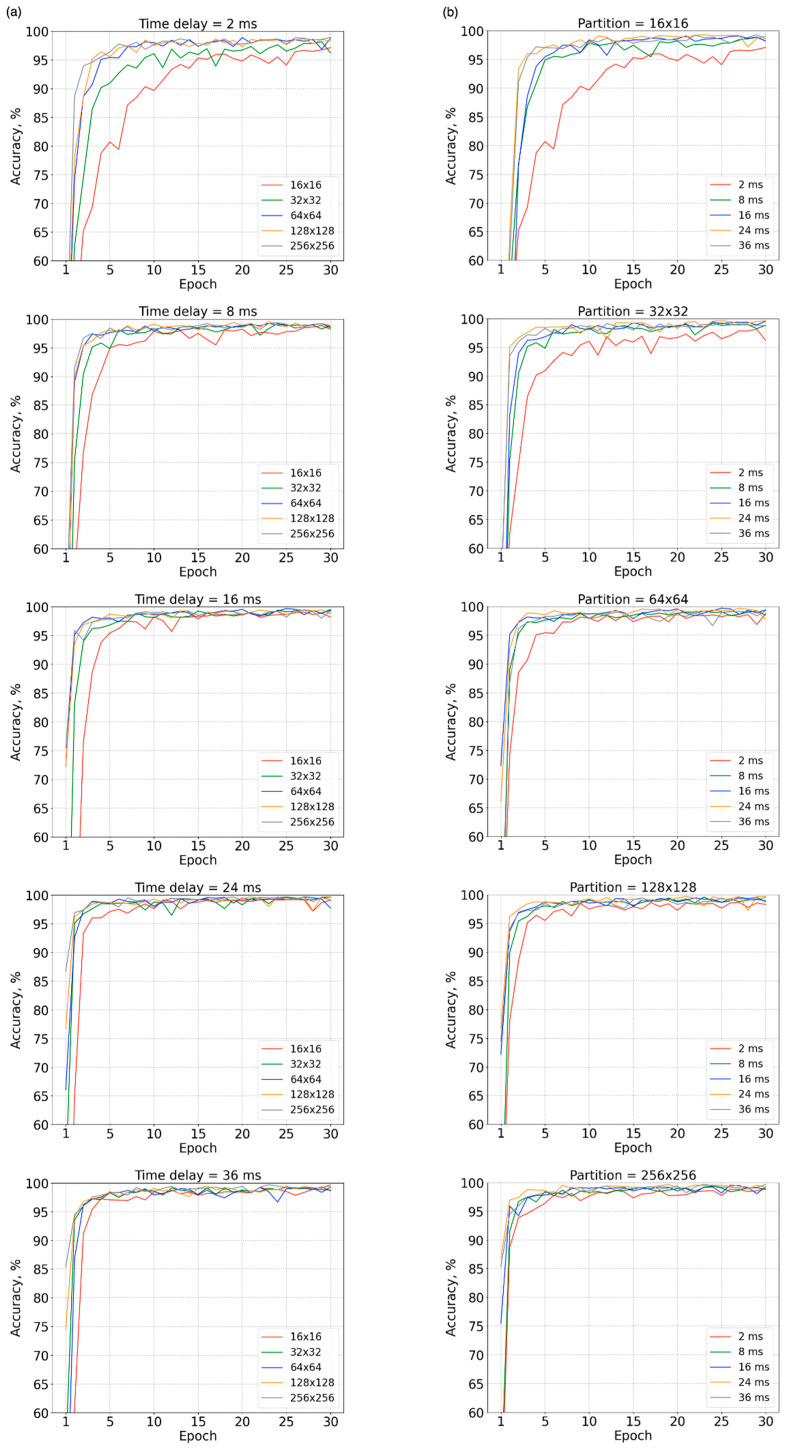
Accuracy of individual identification over the first 30 epochs on the underlying phase portraits (**a**) embedded using five time delays (*τ* = 2, 8, 16, 24, and 36 ms) and (**b**) generated over five grid partitions (16 × 16, 32 × 32, 64 × 64, 128 × 128, and 256 × 256).

**Figure 4 sensors-23-03164-f004:**
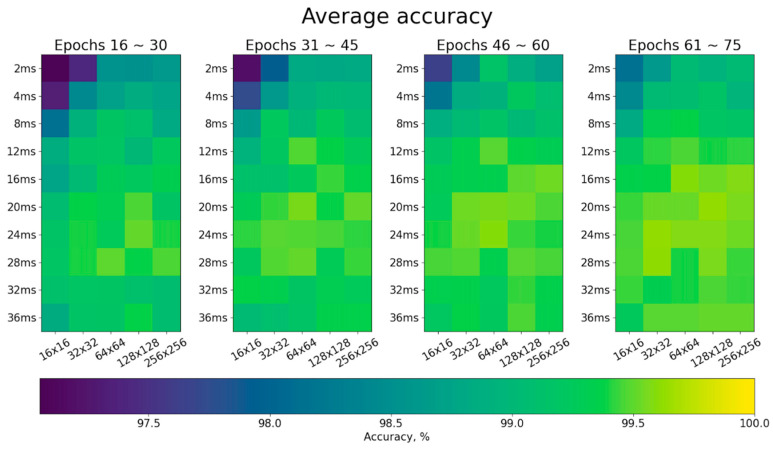
Map of average accuracy of individual identification on the underlying phase portraits using the combinations of five grid partitions (16 × 16, 32 × 32, 64 × 64, 128 × 128, and 256 × 256) and ten time delays (2, 4, 8, 12, 16, 20, 24, 28, 32, and 36 ms) during four training stages (epochs 16~30, 31~45, 46~60, and 61~75).

**Figure 5 sensors-23-03164-f005:**
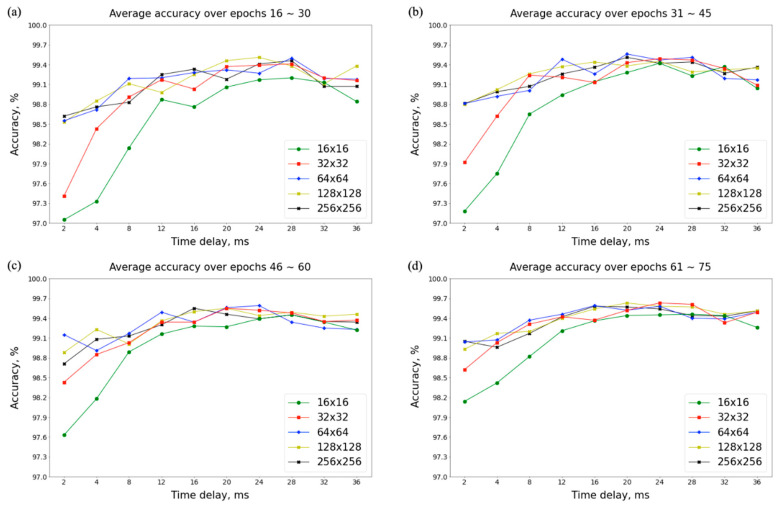
Plot of average accuracies of individual identification on the underlying phase portraits using the combinations of five grid partitions (16 × 16, 32 × 32, 64 × 64, 128 × 128, and 256 × 256) and ten time delays (2, 4, 8, 12, 16, 20, 24, 28, 32, and 36 ms) during four training stages: (**a**) epochs 16~30, (**b**) epochs 31~45, (**c**) epochs 46~60, and (**d**) epochs 61~75.

**Figure 6 sensors-23-03164-f006:**
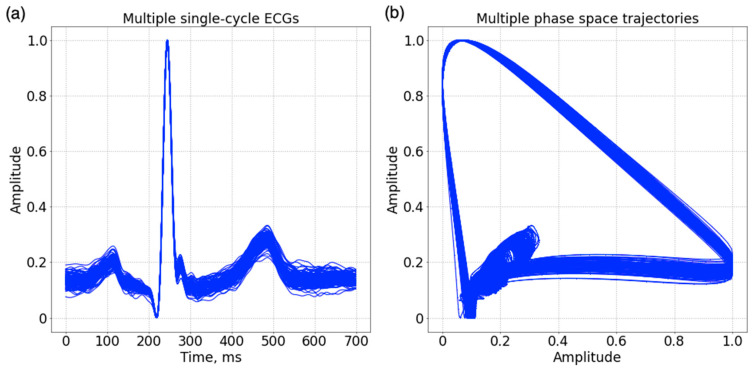
(**a**) Waveforms of 100 consecutive single-beat ECG in a subject. (**b**) The transformed phase space trajectories using a time delay of 20 ms.

**Figure 7 sensors-23-03164-f007:**
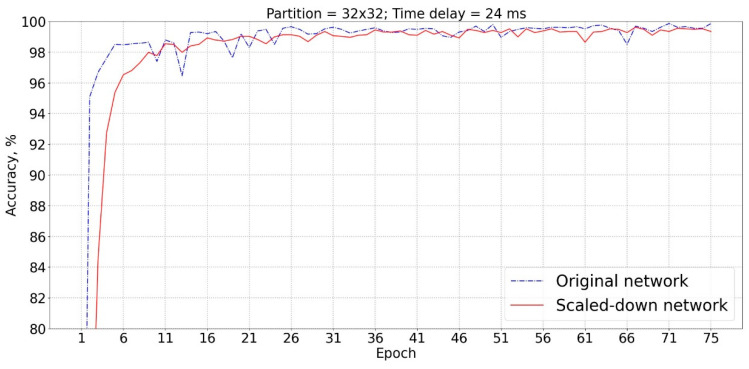
The identification accuracy using the scaled-down network and original network which were designed for portrait images with 32 × 32 and 256 × 256 pixels, respectively, but the portrait images were derived on the basis of phase-space reconstruction over 32 × 32 grid partition.

**Table 1 sensors-23-03164-t001:** The network architecture used as ECG biometric based on ECG phase portrait with 256 × 256 pixels.

	Layer	Input	Output	Kernel	Padding	Stride
Feature extraction	Conv2d	256 × 256 × 1	62 × 62 × 96	12 × 12		4
	Max pooling	62 × 62 × 96	31 × 31 × 96	2 × 2		2
	Conv2d	31 × 31 × 96	31 × 31 × 256	5 × 5	2	1
	Max pooling	31 × 31 × 256	15 × 15 × 256	3 × 3		2
	Conv2d	15 × 15 × 256	15 × 15 × 384	3 × 3	1	1
	Conv2d	15 × 15 × 384	15 × 15 × 384	3 × 3	1	1
	Conv2d	15 × 15 × 256	15 × 15 × 256	3 × 3	1	1
	Max pooling	15 × 15 × 256	7 × 7 × 256	3 × 3		2
Classifier	FCN	12,544	4096			
	Dropout *p* = 0.5					
	FCN	4096	4096			
	Dropout *p* = 0.5					
	FCN	4096	115			

The activation function is chosen as ReLu function in the hidden layers, where the output layer uses softmax function to represent a categorical probability distribution.

**Table 2 sensors-23-03164-t002:** Average accuracy of individual identification on the underlying phase portraits using the combinations of five grid partitions and five time delays during four training stages.

	16 × 16	32 × 32	64 × 64	128 × 128	256 × 256
Epochs 16~30					
τ = 2 ms	97.05	97.41	98.55	98.53	98.62
τ = 8 ms	98.14	98.91	99.19	99.11	98.83
τ = 16 ms	98.76	99.03	99.28	99.25	99.33
τ = 24 ms	99.17	99.39	99.27	99.51	99.41
τ = 36 ms	98.84	99.16	99.18	99.38	99.07
Epochs 31~45					
τ = 2 ms	97.18	97.92	98.81	98.80	98.81
τ = 8 ms	98.65	99.24	99.01	99.26	99.07
τ = 16 ms	99.14	99.13	99.26	99.44	99.36
τ = 24 ms	99.42	99.49	99.47	99.45	99.42
τ = 36 ms	99.04	99.09	99.17	99.35	99.36
Epochs 46~60					
τ = 2 ms	97.63	98.43	99.15	98.88	98.71
τ = 8 ms	98.89	99.03	99.17	99.01	99.13
τ = 16 ms	99.28	99.34	99.34	99.50	99.55
τ = 24 ms	99.39	99.52	99.59	99.44	99.39
τ = 36 ms	99.22	99.37	99.23	99.46	99.34
Epochs 61~75					
τ = 2 ms	98.14	98.62	99.04	98.93	99.05
τ = 8 ms	98.82	99.31	99.37	99.2	99.17
τ = 16 ms	99.36	99.37	99.59	99.54	99.58
τ = 24 ms	99.45	99.63	99.58	99.58	99.54
τ = 36 ms	99.26	99.49	99.49	99.51	99.51

**Table 3 sensors-23-03164-t003:** The scaled-down network used as ECG biometric on the basis of phase portrait with 32 × 32 pixels.

	Layer	Input	Output	Kernel	Padding	Stride
Feature extraction	Conv2d	32 × 32 × 1	28 × 28 × 32	5 × 5		1
	Max pooling	28 × 28 × 32	14 × 14 × 32	2 × 2		2
	Conv2d	14 × 14 × 32	13 × 13 × 32	2 × 2		1
Classifier	FCN	5408	1024			
	Dropout *p* = 0.5					
	FCN	1024	1024			
	Dropout *p* = 0.5					
	FCN	1024	115			

The activation function is chosen as ReLu function in the hidden layers, where the output layer uses softmax function to represent a categorical probability distribution.

**Table 4 sensors-23-03164-t004:** The state of the art in ECG biometric authentication methods using two-dimensional convolutional neural networks.

Authors	Dataset	Subject No.	Input ECG	Transformation	Model	Accuracy, %
Hammad et al. [20]	PTB [31] CYBHi [40]	100 65	single beat	plot of amplitude vs. time	VGG [36]	96.8 97.2
Byeon et al. [21]	PTB, CU [39]	211 100	single beat	continuous wavelet transform	AlexNet [34] GoogleNet [37] ResNet [38]	97.4, 92.3 97.8, 93.1 98.1, 93.2
Ciocoiu and Cleju [22]	UofT [41]	20	single beat	phase-space reconstruction	3-layer CNN	97.2
Kim et al. [24]	MIT-BIH NSRDB [42]	31	multiple beats	plot of amplitude vs. time with beats stacking	ensemble of 2 CNNs and RNN	98.9
Zhang and Zhou [25]	SADB [43]	10	multiple beats	phase-space reconstruction	3-layer CNN	98.8
Present study	PTB	115	single beat	phase-space reconstruction	AlexNet	99.5

All models are constructed using convolutional neural networks (CNN) with or without recurrent neural network (RNN).

## Data Availability

https://physionet.org/content/ptbdb/1.0.0/.

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
