# Peer review of "Convolutional Neural Network for Individual Identification Using Phase Space Reconstruction of Electrocardiogram"

_sensors, 2023, doi:10.3390/s23063164_

Round 1

Reviewer 1 Report

1.      Lines 38-39: «A typical ECG is composed of P, QRS complex, and T waves which are generated by 39 a sequence of atrial depolarization, ventricular depolarization and repolarization. » Why is the U wave excluded?

2.      Lines 58-59: « The other methods transform single-beat ECG to an image by plotting ECG signal on a grayscale image…» What is the ECG signal superimposed on?

3.      Lines 66-70: «However, the employed time delay determines whether the generated phase-space trajectory is sufficiently embedded or not. The density of grid partitions determines whether the transformed phase portrait is fine or coarse. The effects of time delay and grid partition on the PSR-based ECG biometric authentication have not been investigated in previous works». What causes the time delay? Does it occur during the transmission of ECG signals or is it created artificially?

4.      Lines 82-84: «After excluding ECG recordings which had a time duration less than 100 s or were seriously corrupted by noises or artifacts, the recordings from 115 subjects were selected to 84 develop our ECG-biometric model». How did the number of healthy and sick patients change after the exclusion of records?

5.      Line 85: « In this study, we chose lead I ECG as an input to the ECG-biometric model because…». The database used consists of 15 lead ECG signals. Why didn't you use the data from the aVR and aVL sensors?

6.      Paragraph 2.3. Why is the modified AlexNet architecture chosen? Why are others not considered?

7.      Lines 135-136: «Each ECG recording was partitioned into two parts: the former 100 heartbeats were included as training data; the latter 25 heartbeats were used to validate the trained network». Lines 82-84 say that the database contains ECG records longer than 100 seconds, but this is not a certainty that there will be 125 heartbeats. What happens to the records of signals from the database, in which the number of heartbeats is more or less? How was the segmentation of the ECG signal stream into heart beats? How many signals were included in the modeling base after segmentation? It is not clear whether one signal was segmented into 125 signals or into 2 signals?

8.      Lines 151-152: «Its QRS, T and P waves are respectively transformed to a major loop, a secondary loop and a minor loop on each portrait image». Is there a U-wave loop?

9.      Paragraph 3. Present the results of your research in the form of a table.

10.  Figure 6 contradicts lines 136-137.

11.  Paragraph 4. Will the identification be correct if between two different identifications a person has a disease of the cardiovascular system?

12.  Paragraph 4. Why is the proposed method of identification better than the existing ones? Compare your method with known identification methods.

Author Response

Thanks for reviewer’s important comments and suggestions. The manuscript has been revised according to the comments and suggestions. The point-to-point responses are described as follows.

1. Lines 38-39: «A typical ECG is composed of P, QRS complex, and T waves which are generated by a sequence of atrial depolarization, ventricular depolarization and repolarization. » Why is the U wave excluded?

Re: Thanks for reviewer’s suggestion. We added U wave to the composition of ECG. Please refer to line 38.

2. Lines 58-59: « The other methods transform single-beat ECG to an image by plotting ECG signal on a grayscale image…» What is the ECG signal superimposed on?

Re: Thanks for remind. The ECG signal is superimposed on a two-dimensional image. To make the statement clear, we revised it as “The other methods transform the one-dimensional ECG signals to two-dimensional (2D) images by plotting each ECG beat as individual grayscale image (time-amplitude representation), decomposing each ECG beat on various scales using continuous wavelet transform (time-scale representation), and embedding each ECG beat onto a 2D phase-space image using a time-delay technique.” Please refer to lines 67-71.

3. Lines 66-70: «However, the employed time delay determines whether the generated phase-space trajectory is sufficiently embedded or not. The density of grid partitions determines whether the transformed phase portrait is fine or coarse. The effects of time delay and grid partition on the PSR-based ECG biometric authentication have not been investigated in previous works». What causes the time delay? Does it occur during the transmission of ECG signals or is it created artificially?

Re: Thanks for important comments. The time delay is empirically determined. Therefore, we added a paragraph to explain the time-delay method: “Phase space reconstruction transforms a 1D ECG signal to a 2D phase-space trajectory by a plot of each ECG sample versus its respective sample after a fixed time delay. Thus, the temporal P, QRS and T waves are mapped to specific spatial loops in phase space without the need of characteristic wave detections. The employed time delay determines whether the reconstructed phase-space trajectory is sufficiently embedded or not. However, the time delay is empirically determined without a study for the effect of using various time delays on the accuracy of the ECG biometric authentication.” Please refer to lines 75-82.

4. Lines 82-84: «After excluding ECG recordings which had a time duration less than 100 s or were seriously corrupted by noises or artifacts, the recordings from 115 subjects were selected to develop our ECG-biometric model». How did the number of healthy and sick patients change after the exclusion of records?

Re: A total of 115 ECG recordings covering 17 healthy subjects, 81 subjects with myocardial infarction, and 15 subjects with cardiomyopathy or heart failure were selected to develop our ECG-biometric model. Please refer to lines 105-108.

5. Line 85: « In this study, we chose lead I ECG as an input to the ECG-biometric model because…». The database used consists of 15 lead ECG signals. Why didn't you use the data from the aVR and aVL sensors?

Re: We revised the description to state why we selected lead I ECG as the input to the ECG-biometric model:

“Compared to lead augmented vector left/right (aVL/aVR) that is electrical voltage difference between left/right arm electrode and a combination of right/left arm electrode and left leg electrode, lead I is defined as voltage difference between the electrodes on left and right arms, which can also be obtained through the fingers of both hands and is the most practical measurement for ECG-biometric authentication. Therefore, we chose lead I ECG as an input to the ECG-biometric model in this study.” Please refer to lines 109-114.

6. Paragraph 2.3. Why is the modified AlexNet architecture chosen? Why are others not considered?

Re: Thanks for important comments. As the comparison in a newly added table (Table 4), two studies use three-layer CNN to extract discernible features from ECG images; two studies and the present study use deep neural networks. As shown in a study by Byeon et al., the accuracy of individual identification was slightly increased as the network went deeper in sequence of AlexNet, GoogleNet, and ResNet. Although it is not easy to compare the accuracy of individual identification among various studies because of various data selections in different datasets, the proposed architecture on the basis of the AlexNet and phase space reconstruction could achieve high accuracy in individual identification. The effect of network depth and architecture on individual identification over the ECG phase-space features can be further studied in future work. Please refer to lines 304-316.

7. I Lines 135-136: «Each ECG recording was partitioned into two parts: the former 100 heartbeats were included as training data; the latter 25 heartbeats were used to validate the trained network». Lines 82-84 say that the database contains ECG records longer than 100 seconds, but this is not a certainty that there will be 125 heartbeats. What happens to the records of signals from the database, in which the number of heartbeats is more or less? How was the segmentation of the ECG signal stream into heart beats? How many signals were included in the modeling base after segmentation? It is not clear whether one signal was segmented into 125 signals or into 2 signals?

Re: The responses to reviewer’s comments are described as the following.

  • All included ECG records had more than 125 heartbeats. Therefore, we revised the description of ECG recording selection as “ECG recordings which had a time duration less than 100 s or were seriously corrupted by noises or artifacts were excluded, so that the included recordings contained more than 125 heart beats with less noise contaminations for network training and testing.” Please refer to lines 102-105.
  • The segmentation of individual ECG signal into heartbeats was performed by the following steps: QRS complex was detected using a Python ECG QRS detector based on the Pan-Tompkins algorithm; each detected QRS was further verified to exclude wrong detections and ventricular premature contraction beats; for each inclusive heartbeat, the preceding 35% and the succeeding 65% samples around QRS fiducial point in a cardiac cycle were extracted. Please refer to lines 121-128.
  • After the aforementioned processing, 125 available heartbeats could be extracted for network training and testing. Please refer to lines 129-130.

8. Lines 151-152: «Its QRS, T and P waves are respectively transformed to a major loop, a secondary loop and a minor loop on each portrait image». Is there a U-wave loop?

Re: Since U-wave is not apparent in the single-beat ECG, no distinct loop is observed. We revised this statement as “Its QRS, T and P waves are respectively transformed to a major loop, a secondary loop and a minor loop on each portrait image but no visible U-related loops are observed because of unapparent U wave in Fig 1.” Please refer to lines 183-185.

9. Paragraph 3. Present the results of your research in the form of a table.

Re: We added Table 2 to list the average accuracies of individual identification on the underlying phase portraits using the combinations of five grid partitions (16´16, 32´32, 64´64, 128´128, and 256´256) and five time delays (2, 8, 16, 24, and 36 ms) during four training stages (epochs 16 ~ 30, 31 ~ 45, 46 ~ 60, and 61 ~ 75). Please refer to line 217 and lines 232-234.

10. Figure 6 contradicts lines 136-137.

Re: Thanks for reminding. We emphasize the training of CNN to learn individual patterns and cover loop variations:

In this study, we used 100 consecutive single-beat phase portraits from each individual to train the network. The existence of loop variations in the training data also provided a kind of data augmentation to increase the robustness of CNN to the loop variation. Please refer to lines 249-252.

11. Paragraph 4. Will the identification be correct if between two different identifications a person has a disease of the cardiovascular system?

Re: Thanks for great comments. Pathological heartbeats caused by extra-systolic ventricular contractions are commonly different from the healthy heartbeats in the same individual, so they cannot be recognized as individual’s pattern by the trained networks on the basis of healthy heartbeats. Fortunately, extra-systolic heartbeats commonly occur intermittently or interlacedly (bigeminy or trigeminy). There are still healthy heartbeats that can be recognized by the trained network. That’s why most ECG-biometric authentication methods select healthy heartbeats to develop their biometric networks. We added a new paragraph in discussion to talk over this issue. Please refer to lines 297-303.

12. Paragraph 4. Why is the proposed method of identification better than the existing ones? Compare your method with known identification methods.

Re: Thanks for great suggestion. We added Table 4 to show the state of the art in ECG-biometric authentication methods using two-dimensional convolutional neural networks. Although it is not easy to compare the accuracy of individual identification among various studies because of various data selections in different datasets, the proposed architecture on the basis of the AlexNet and phase space reconstruction could achieve high accuracy in individual identification. Please refer to lines 304-320.

Reviewer 2 Report

This manuscript developed a PSR-based CNN for ECG biometric authentication and achieved a higher accuracy. The effects of time delay and grid partition have been investigated based on a population of 115 subjects selected from the PTB Diagnostic ECG Database. This study is helpful to reduce the network size and training time required for individual identification. I think the manuscript meets the criteria for publication on Sensors.

Author Response

This manuscript developed a PSR-based CNN for ECG biometric authentication and achieved a higher accuracy. The effects of time delay and grid partition have been investigated based on a population of 115 subjects selected from the PTB Diagnostic ECG Database. This study is helpful to reduce the network size and training time required for individual identification. I think the manuscript meets the criteria for publication on Sensors.

Re: Thanks for reviewer’s recommendation.

Reviewer 3 Report

The paper presents an interesting work dealing with the person identification using phase space reconstruction of electrocardiogram by means of neural networks.

The paper is well structured and the content well balanced.

However, several issues that the Authors have to solve prior to the "Acceptance/Rejection" decision of this manuscript are listed bellow:

1. The introduction section should be improved in order to better highlight the importance of having good quality of the acquired ECG signals from individuals (prior to apply any statistical learning algorithm for biometric applications)

For example, methods already published in this field are available in literature. For example, the following paper may be cited:

N. Adochiei, V. David, F. Adochiei and I. Tudosa, "ECG waves and features extraction using Wavelet Multi-Resolution Analysis," 2011 E-Health and Bioengineering Conference (EHB), Iasi, Romania, 2011, pp. 1-4.

2. At the end of the Introduction Sectio, a paragraph describing the paper structure will be welcome. 

3. Section II. Subsection 2.1. There is a proposition "Each signal is digitized at 1 kHz." this should be changed to "Each signal is digitized at 1 kSa/s."  - kSa/s = kilo Samples per second. And moreover, which is the sample length in terms of bits? 

4. Section II. Subsection 2.2. There is no discussion about the importance of the quality of ECG signals. The Authors have to add few sentences describing that the ECG signal have to preserve a high Signal-to-Noise Ration prior to be applied into a CNN algorithm.

Best Regards,

The Reviewer 

Author Response

The paper presents an interesting work dealing with the person identification using phase space reconstruction of electrocardiogram by means of neural networks.

The paper is well structured and the content well balanced.

However, several issues that the Authors have to solve prior to the "Acceptance/Rejection" decision of this manuscript are listed below:

Re: Thanks for reviewer’s important comments and suggestions. The manuscript has been revised according to the comments and suggestions. The point-to-point responses are described as follows.

1. The introduction section should be improved in order to better highlight the importance of having good quality of the acquired ECG signals from individuals (prior to apply any statistical learning algorithm for biometric applications)

For example, methods already published in this field are available in literature. For example, the following paper may be cited:

N. Adochiei, V. David, F. Adochiei and I. Tudosa, "ECG waves and features extraction using Wavelet Multi-Resolution Analysis," 2011 E-Health and Bioengineering Conference (EHB), Iasi, Romania, 2011, pp. 1-4.

Re: Thanks for reviewer’s suggestion. We added the importance of good quality of the acquired ECG signals for individual identification to two paragraphs in introduction:

  • The good quality of the acquired ECG signals is essential, especially in the early ECG biometric methods. The identifications of small-amplitude characteristics like P, Q, and S waves are commonly affected by the presence of noise contaminations or when the characteristics are difficult to discern. Please refer to lines 45-48.
  • Even though the waveform-based methods do not need the detections of the small-amplitude waves, QRS detection is still needed for the segmentation of a cardiac cycle for individual identification. Digital filtering is commonly used to attenuate low-frequency baseline wandering and high-frequency noises. Advanced methods using multi-resolution wavelet analysis and Kalman filtering are also proposed to enhance ECG waveforms for QRS detection and extract ECG features. Please refer to lines 53-59.

2. At the end of the Introduction Section, a paragraph describing the paper structure will be welcome. 

Re: Thanks for great suggestion. We added a paragraph to describe paper structure:

The rest of this paper is organized as follows. The method section describes ECG biometric data, phase space reconstruction of ECG, and ECG biometric network. In the result section we present the accuracies of individual identification on the basis of phase-space features using various time delays and various-density grid partitions. In the discussion section we talk over beat-to-beat variation, the effects of time delay and grid partition on identification accuracy, and the scaled-down of the proposed network. Please refer to lines 89-94.

3. Section II. Subsection 2.1. There is a proposition "Each signal is digitized at 1 kHz." this should be changed to "Each signal is digitized at 1 kSa/s."  - kSa/s = kilo Samples per second. And moreover, which is the sample length in terms of bits?

Re: Thanks for suggestion. We revised the sampling by “Each signal is digitized at 1 kSa/s (kilo samples per second) with a resolution of 16 bits.” Please also refer to lines 101-102.

4. Section II. Subsection 2.2. There is no discussion about the importance of the quality of ECG signals. The Authors have to add few sentences describing that the ECG signal have to preserve a high Signal-to-Noise Ratio prior to be applied into a CNN algorithm.

Re: Thanks for remind. We added a statement to address the importance of ECG quality:

Since the good quality of the ECG signals is important for recognizing individuals’ ECG patterns, we employed digital filtering to keep a high signal-to-noise ratio in the processed ECGs prior to the application to the ECG-biometric model. Please refer to lines 116-118.

Reviewer 4 Report

Authors should explain in more detail the rationale behind this work and potential applications as well as comparison with the work of others.

How are their methods linked to a specific electrophyisological ECG pattern (temporal and spatial). Why is the proposed method relevant for ECG classification? All of these points have to be elaborated within introduction and discussion.

Author Response

Thanks for reviewer’s important comments and suggestions. The manuscript has been revised according to the comments and suggestions. The point-to-point responses are described as follows.

1. Authors should explain in more detail the rationale behind this work and potential applications as well as comparison with the work of others.

Re: Thanks for great suggestion. The responses to reviewer’s comments are described as the following.

We added several statements to explain in more detail the rationale behind our work and potential applications:

  • A typical ECG is composed of P, QRS complex, T, and U waves which are generated by a sequence of atrial depolarization, ventricular depolarization and repolarization. Because each person has respective cardiac conduction system and projection of the generated electrical activity on body surface, the waveform of the recorded ECG varies from person to person. Please refer to lines 38-42.
  • The employed time delay determines whether the reconstructed phase-space trajectory is sufficiently embedded or not. However, the time delay is empirically determined without a study for the effect of using various time delays on the accuracy of the ECG biometric authentication. In addition, the density of grid partitions determines whether the transformed phase portrait is fine or coarse. The effect of grid partition on the accuracy is also not reported in previous works. Please refer to lines 79-84.
  • In summary, temporal-to-spatial mapping through PSR directly produces 2D images with disclosed ECG characteristics, enabling an efficient learning of individual fingerprints by the 2D CNN. The use of the scaled-down CNN with appropriate time delay largely reduces the complexity of the model with a maintenance of identification performance, which is beneficial for practical implementation of the ECG-biometric on authentication systems. Please refer to lines 330-335.

We also added Table 4 to show the state of the art in ECG-biometric authentication methods using two-dimensional convolutional neural networks. Although it is not easy to compare the accuracy of individual identification among various studies because of various data selections in different datasets, the proposed architecture on the basis of the AlexNet and phase space reconstruction could achieve high accuracy in individual identification. Please refer to lines 304-320.

2. How are their methods linked to a specific electrophyisological ECG pattern (temporal and spatial). Why is the proposed method relevant for ECG classification? All of these points have to be elaborated within introduction and discussion.

Re: Thanks for comments. The responses to reviewer’s comments are described as the following.

  • The proposed method is linked to the transformation of temporal ECG patterns to spatial loops in phase space. Therefore, we revised the description of phase space reconstruction in introduction: “Phase space reconstruction transforms an ECG signal to a phase-space trajectory by a plot of each ECG sample versus its respective sample after a fixed time delay. Thus, the temporal patterns of the ECG signal (P, QRS and T waves) are mapped to specific spatial loops in phase space without the need of characteristic wave detections.” Please also refer to lines 75-79.
  • We also emphasized this in discussion: “PSR coverts morphological characteristics (temporal patterns) in an ECG signal into phase-space loops (spatial patterns). The converted spatial patterns are different among individuals, leading to good performances in ECG biometrics. In particular, each spatial pattern has specific locations in the phase space, allowing 2D CNN to efficiently capture the phase-space fingerprints.“ Please also refer to lines 257-261.
  • The proposed method is not relevant for ECG classification. To avoid causing confusion to the readers, we remove the description of “PSR has demonstrated a good performance in ventricular premature contraction detection” from the revised manuscript.

Round 2

Reviewer 1 Report

All comments have been addressed.

Author Response

Comments and Suggestions for Authors

All comments have been addressed.

Re: Thanks for reviewer’s kindly comments and recommendation.

Reviewer 4 Report

The paper would benefit from better explaination of the time delay parameter. What is the actual meaning behind this parameter and what is the electrophysiological significance of the "time delay" since obviosly results vary significantly depending on this. This should be further explained.

Also, the aims (or research questions) and novelty of the paper should be very precisely defined at the end of the introduction. 

Author Response

Comments and Suggestions for Authors

1. The paper would benefit from better explanation of the time delay parameter. What is the actual meaning behind this parameter and what is the electrophysiological significance of the "time delay" since obviously results vary significantly depending on this. This should be further explained.

Re: Thanks for great suggestion. We added a new paragraph to describe the use of time delay to reconstruct phase space of ECG with the purposes of investigations of the nonlinear dynamics disclosed by ventricular fibrillation, QRS detection and ventricular arrhythmia classifications, and ECG-biometric authentication based on the morphology of the phase-space trajectory:

Phase space reconstruction (PSR) uses a time-delay technique to reconstruct the phase-space trajectory of a signal. The signal is embedded into a multi-dimensional phase space by a plot of each sample x(t) versus its respective samples after fixed time delays x(t+τ), x(t+2τ), …, x(t+(m-1)τ), which was proposed initially for the purpose of disclosing nonlinear dynamics of the signal on the basis of chaos physics [26]. PSR was firstly used to analyze ECG for examining whether ventricular fibrillation is an instance of deterministic chaos [27]. Moreover, the reconstructed phase portraits reveal various morphologies associated with normal and pathological ECGs. Thus, several studies employed the PSR with m=2 to detect QRS complex [28], recognize ventricular extrasystoles [29], and classify the type of ventricular arrhythmia [30] because the 2D reconstruction displays a concise phase space trajectory and lends itself more readily to feature extraction. In addition, PSR transforms the temporal patterns of the ECG signal (P, QRS and T waves) to specific spatial loops in phase space without the need of characteristic wave detections, extending its application to individual identification [12,13,22,24].

Please refer to lines 75-89.

2. Also, the aims (or research questions) and novelty of the paper should be very precisely defined at the end of the introduction. 

Re: Thanks again for reviewer’s kindly remind. We revised the motivation, aim, novelty of the paper at the end of introduction as follows:

The value of time delay determines the morphology of the reconstructed phase-space trajectory and whether the ECG characteristics are sufficiently expanded in the phase space. However, the time delay is empirically determined in the majority of studies without an investigation to explore the effect of various time delays on the accuracy of the ECG biometric authentication. On the other hand, the phase space is partitioned by a series of intersecting vertical and horizontal lines. The grid partition generates a 2D image array to record the absence or presence of the reconstructed trajectory in each partitioned square. The density of the grid partition determines whether the generated portrait image is fine-grained or coarse-grained, which may affect the accuracy of individual identification but it has not been reported in previous works. In this study, we propose a novel ECG-biometric model which utilizes a deep CNN to learn and identify individuals’ patterns over the reconstructed phase portrait. We also investigate the aforementioned effects on the accuracy of the proposed ECG-biometric model using the underlying combinations of various time delays (τ = 2, 8, 16, 24, and 36 ms) and various-density grid partitions (16x16, 32x32, 64x64, 128x128, and 256x256) on the basis of the data from the PTB Diagnostic ECG Database, with a purpose of determining an optimal time delay and a lower-complexity model with the maintenance of identification performance.

Please refer to lines 90-107.